

# BlockDroid: detection of Android malware from images using lightweight convolutional neural network models with ensemble learning and blockchain for mobile devices

Emre Şafak[1,2], İbrahim Alper Doğru[2], Necaattin Barışçı[2] and İsmail Atacak[2]

[1] Department of Information and Communication Technologies, HAVELSAN, Ankara, Turkey
[2] IoTLab, Department of Computer Engineering, Gazi University, Ankara, Turkey

Corresponding author
Emre Şafak, emresfk2@gmail.com

## ABSTRACT

Due to the increase in the volume and diversity of malware targeting Android systems, research on detecting this harmful software is steadily growing. Traditional malware detection studies require significant human intervention and resource consumption to analyze all malware files. Moreover, malware developers have developed polymorphism and code obfuscation techniques to evade traditional signature-based detection approaches used by antivirus companies. Consequently, traditional methods have become increasingly inadequate for malware detection. So far, many machine learning methods have been successfully applied to address the issue of malware detection. Recent efforts in this area have turned to deep learning methods. Because these methods can automatically extract meaningful features from data and efficiently learn complex relationships, they can achieve better performance in malware detection as well as in solving many other problems. This article presents BlockDroid, an approach that combines convolutional neural network (CNN) models, ensemble learning, and blockchain technology to increase the accuracy and efficiency of malware detection for mobile devices. By converting Android DEX files into image data, BlockDroid leverages the superior image analysis capabilities of CNN models to discern patterns indicative of malware. The CICMalDroid 2020 dataset, comprising 13,077 applications, was utilized to create a balanced dataset of 3,590 images, with an equal number of benign and malware instances. The proposed detection system was developed using lightweight models, including EfficientNetB0, MobileNetV2, and a custom model as CNN models. Experimental studies were conducted by applying both individual models and the proposed BlockDroid system to our dataset. The empirical results illustrate that BlockDroid surpasses the performance of the individual models, demonstrating a substantial accuracy rate of 97.38%. Uniquely, BlockDroid integrates blockchain technology to record the predictions made by the malware detection model, thereby eliminating the need for re-analysis of previously evaluated applications and ensuring more efficient resource utilization. Our approach offers a promising and innovative strategy for effective and efficient Android malware detection.

# INTRODUCTION

Malicious software, or malware, is a malevolent code that harms computer devices and networks, and steals sensitive information, often without user's consent. Common variants include spyware, ransomware, viruses, worms, trojans, adware, or any malicious code that infiltrates a computer (*Razgallah et al., 2021*). Malware attacks are among the most common threats identified for Android. Many researchers have provided various definitions for malware based on the damage they cause. Malware can be defined as a malicious application containing a piece of malicious code designed to gain unauthorized access and violate the three main principles of security: confidentiality, integrity, and availability (*Senanayake, Kalutarage & Al-Kadri, 2021*). According to a market share report on smartphone operating systems in 2022, Android's market share has risen to 83.8%. This significant share has prompted the development of Android malware. The Chianxin Threat Intelligence Center's report on Android malware revealed that approximately 2.3 million new malicious software were detected on mobile terminals, with an average of about 6,301 new mobile phone malware blocked per day. Malicious activities identified include malicious fee deductions at a rate of 34.9%. Malicious software has also become a significant problem for Android devices, accounting for 24.2% of resource consumption, 22.8% of fake behavior, 12.3% of privacy breaches, 4.3% of deception and fraud, and only 1.5% of remote control (*Shatnawi, Yassen & Yateem, 2022*). Therefore, to combat this issue, Android malware detection efforts are increasing. These efforts are generally divided into three categories: static, dynamic, and hybrid analysis. Static analysis extracts features from Android Package (APK) files without executing them (*Arslan, Doğru & Barişçi, 2019*), proving effective in identifying malware through permissions, APIs, and Android-related intents, methods, and components. However, it may struggle when encountering complex code structures or Technologies (*Sharma & Arora, 2024*). Dynamic analysis involves monitoring the application in a protected environment, and collecting and analyzing behavioral information, but can be time-consuming. Hybrid analysis is an approach that combines both static and dynamic analysis methods (*Elayan & Mustafa, 2021*). It has the advantages of both methods and can capture different characteristics of malware, resulting in more accurate predictions of malware detection. The long analysis time and high resource consumption are the main disadvantages of this approach. Recent studies have shown that machine learning and deep learning can effectively detect malware. In particular, deep learning methods, which have proven effective in areas such as image recognition, natural language processing, signal processing, and speech processing, natural language processing, signal processing, and speech processing, have also achieved notable achievements in Android malware detection (*Güngör et al., 2022*). In these studies, most deep learning methods have been applied to

static or dynamic Android malware data with extracted features. Nevertheless, the number of studies conducted on image-based data is relatively low compared to the current studies. They generally focus on applying pure or hybrid deep learning methods to images converted from Android source files and data sources.

Additionally, there are different approaches to detecting Android malware through QR code-like images based on selected permission information (*Kılıç, Doğru & Toklu, 2024*). Converting Android DEX files into images for analysis using deep learning techniques enables the more effective and faster detection of malware that traditional methods may miss. This approach aims to encode every piece of information within the DEX file as an image, allowing for more accurate and efficient malware analysis, and potentially paving the way for a new era in cybersecurity.

In this study, a CNN-based ensemble approach combined with blockchain was developed as a cost-effective and secure malicious software detection system called BlockDroid. In this structure, the ensemble approach was implemented to improve the detection of malware in Android systems, while blockchain technology was utilized to securely store the predictions. Since images converted from DEX files are used in Android malware detection, CNN-based models known to yield successful results in image analysis were employed in this approach. Lightweight methods, including EfficientNetB0, MobileNetV2, and a custom CNN, were preferred to ensure the system's feasibility in mobile device environments. The contributions of the proposed system to the literature can be summarized in the following three points.

- The accuracy rate in Android malware detection has been increased by employing the stacked ensemble learning method together with MobileNetV2, EfficientNetB0, and a customized model.
- A custom model for detecting Android malware has been developed, utilizing lightweight models in conjunction with ensemble learning to achieve high accuracy. The decisions made by the malware detection model are securely stored using blockchain technology, and the model is prevented from being rerun for the same application. Additionally, the malware detection model is executed on nodes in the blockchain, ensuring immutability and security.
- By integrating blockchain technology, it is ensured that model decisions are stored in an immutable manner, adding a layer of trust and security to the decision-making process.
- By utilizing lightweight convolutional neural network (CNN) models, the malware detection model is deployed on mobile devices, reducing computational costs and enabling user control directly on the mobile device. Moreover, potential bottlenecks on central servers due to an increase in the number of users are prevented.

The other sections of the study are organized as follows: The second section provides an overview of current studies in the literature. The third section describes in detail the materials and methods used in the study. The fourth section compares and discusses the experimental results of the proposed method with the results of other methods used in the

study and those reported in the literature. In the final section, the results are evaluated overall, and potential directions for future studies are presented.

## LITERATURE REVIEW

A considerable number of studies have been conducted in malware detection. In this section, a review study is presented regarding machine learning, deep learning, and hybrid approaches related to the study provided here.

Since signature-based solutions for detecting malware on mobile devices cannot analyze the entire application comprehensively, machine learning-based behavioral analysis solutions have begun to be developed. *Bose et al. (2008)* proposed a new behavioral detection framework to identify mobile worms, viruses, and trojans instead of relying on signature-based methods. The malicious behaviors of malware were distinguished from normal application behaviors using a classifier based on support vector machines (SVMs). Evaluations on both simulated and real-world malware samples showed that behavioral detection could identify existing mobile viruses and worms with over 96% accuracy (*Bose et al., 2008*). Machine learning-based behavioral analysis solutions, which achieved promising results on mobile devices, were later adapted for the widely used Android operating system. One of the earliest studies applying machine learning for Android malware detection was conducted by *Sahs & Han (2012)*. In their work, 2,081 benign and 91 malicious Android applications were analyzed using permissions and control flow graphs for classification. Their experiments achieved an 85% accuracy rate using SVMs (*Sahs & Han, 2012*). To improve upon the results of traditional machine learning, deep learning algorithms—known for their strong performance in classification tasks—began to be employed. One of the first studies using deep learning for Android malware detection was conducted by *Yuan et al. (2014)*. In this study, 200 features extracted from 500 Android applications (250 malicious and 250 benign) were used for classification, achieving a 96.5% accuracy rate, outperforming traditional machine learning algorithms.

Research in the field of Android malware detection has significantly progressed with the use of machine learning (ML), deep learning (DL), and hybrid approaches. These studies have focused on static analysis of application information and converting application files into images for malware detection. However, due to the high computational costs of the developed methods, they are generally not suitable for use on mobile platforms.

This section reviews studies on Android malware detection that utilize application data analysis, image-based techniques, and detection methods for mobile devices. The literature review focuses on Android malware detection studies employing static and image-based analysis. As inclusion criteria, peer-reviewed studies, research proposing concrete models, and studies tested on publicly available datasets were determined. Among the exclusion criteria are review articles only, non-peer-reviewed publications, studies not targeting Android malware, and research not using machine learning or deep learning-based classification approaches.

In studies involving static analysis of application information, features such as permissions, APIs, and Android-related intents, methods, and components are used to detect Android malware through machine learning and deep learning techniques. These

studies typically require server-level computational power, which does not make them suitable for direct deployment on mobile devices. There are also no developments aimed at preventing repetitive operation of the model or ensuring model security. *Sönmez, Salman & Dener (2021)* used the CICMalDroid2020 dataset to train and test machine learning algorithms including K-nearest neighbors, naive Bayes, J48, and random tree with 470 features. The K-nearest neighbors algorithm achieved the highest accuracy at 91.45% (*Sönmez, Salman & Dener, 2021*). However, the study focused solely on permissions and API calls, limiting its ability to detect anomalies in code structure. While classical machine learning methods were used, deep learning models were not evaluated. The accuracy rate was also lower than similar studies using the same dataset. *Azad et al. (2022)* a particle swarm optimization method and a neural network consisting of two hidden layers were employed. The utilized CICAndMal2017 dataset includes network traffic, memory dumps, logs, permissions, API related searches, and various phone statistics collected from various Android applications. In the feature extraction process, spiral-based particle swarm optimization (PSO), which employs a learning algorithm to assess the utility of a feature, was employed. Among the methods used to observe the classification impact of feature selection, the PSO method exhibited the best performance in the deep learning classifier with an accuracy rate of 83.6% and an F-score of 82.5% when reducing the data size by 300. The obtained results are limited when compared to other methods offering higher accuracy rates (*Azad et al., 2022*). *Azad et al. (2022)* employed PSO and neural networks but achieved only 83.6% accuracy, revealing the challenge of balancing feature reduction and performance. Furthermore, no comparison has been made with studies conducted using the CICAndMal2017 dataset. *Faghihi, Zulkernine & Ding (2023)* ensured the interpretability of the model's decision for Android malware detection. They introduced a novel approach named Android Interpretable Malware (AIM) for detecting malicious software. AIM integrates hybrid analysis, neural networks, attention mechanism, and a novel class modeling approach. Leveraging the attention mechanism, this method can discern the predicted label of an application (benign or malicious), along with the classes engaged in malicious activities, API calls, and permissions. The proposed AIM detector attained an accuracy rate of 98.91% (*Faghihi, Zulkernine & Ding, 2023*). The performance of the proposed method was measured using the dataset prepared for the study. However, its performance has not been validated using one of the commonly used datasets in the literature. *Kong et al. (2022)* proposed a new method named Feature-Centric Siamese Convolutional Neural Network (FCSCNN) for Android malware detection. Initially, permissions and API calls were extracted from the Manifest.xml and smali files of both benign and malicious applications within the dataset. In the research, features were categorized into uncertain, malicious, and benign groups. Subsequently, Siamese CNNs were employed to detect malicious applications. The proposed FCSCNN method achieved a higher accuracy rate of 98.07% compared to previous studies *Kong et al. (2022)*. While it is claimed that the proposed method is more effective against new malware, this has not been experimentally proven. Instead of using an existing dataset, a new dataset was prepared; however, the performance of the proposed method has not been evaluated with another dataset. In the study, a new method named WHGDroid (Weighted Heterogeneous

Graph, WHG) was proposed for Android malware detection. WHGDroid consists of four main components: feature extractor and weight assigner, WHG and metapath constructor, node embedding, and malicious software detection. The proposed WHGDroid method was tested against evolving (evolution) and never-before-seen (zero-day) malicious software scenarios. WHGDroid outperformed baseline graph-based methods in evolution (92.50%) and zero-day (97.83%) scenarios (*Huang et al., 2023*). Feature extraction is performed manually and takes approximately 7.8 s. Therefore, the proposed method is limited in use, as it will result in longer training times for large datasets. *Zhu et al. (2023a)* proposed MSerNetDroid model consisting of three main components: multi-static feature extraction, feature vector formation and reshaping, and feature learning-based classification. Permissions and hardware features were extracted from the manifest file, and API calls were extracted from the DEX files using Androguard. Based on prior research, 145 permissions, 94 sensitive API calls, and 85 hardware features were employed in the training process. Experimental results demonstrated that the proposed MSerNetDroid framework achieved a higher accuracy rate of 96.48% compared to previous studies (*Zhu et al., 2023a*). The dataset was prepared within the scope of the study, and the performance of the proposed method on malicious software it has not previously encountered, using a different dataset, has not been evaluated. *Atacak, Kılıç & Doğru (2022)* using the CICMalDroid2020 dataset, fuzzy logic has been applied together with convolutional neural network architecture for feature extraction and decision-making processes. Their approach reduced feature dimensionality by extracting features from permission information using a small number of filters and convolutional layers. The proposed method attained an accuracy of 94.6% and an F1-score of 94.6% on the CICMalDroid2020 dataset (*Atacak, Kılıç & Doğru, 2022*). In the proposed method, analyzing only permission information may result in the inability to detect malware applications caused by other features beyond permissions. *Padmavathi, Shanmugapriya & Roshni (2022)* leveraging the CICMalDroid2020 dataset, conducted experiments with six unsupervised machine learning models: K-means, K-nearest neighbors (KNN), Density-Based Spatial Clustering of Applications with Noise (DBSCAN), Ordering Points To Identify the Clustering Structure (OPTICS), hierarchical, and spectral clustering, to detect mobile malicious software. Through a comparative analysis of these models, it was found that the K-means clustering algorithm achieved the highest accuracy of 88% for detecting mobile malicious software (*Padmavathi, Shanmugapriya & Roshni, 2022*). Classical machine learning algorithms have been applied, but deep learning models have not been explored. The accuracy achieved is lower compared to similar studies using the same dataset. *Kou et al. (2023)* using the CICMalDroid2020 dataset, semi-supervised, continuously learning malware detection model based on Transformer has been proposed. The proposed model introduces a feature memory replay algorithm and a pseudo-labeling algorithm. Multilayer Perceptron was employed for classifying malicious software. This model attained an F1 score of 85.49% on the CICMalDroid2020 dataset (*Kou et al., 2023*). The performance achieved has been lower compared to other studies in the literature that use the same dataset. *Gu, Hongyan & Hou (2024)* proposed a novel malware detection framework called MFEMDroid, which combines multi-feature analysis and ensemble

modeling. The framework incorporates a composite model based on SENet, ResNet, and SEResNet architectures. To comprehensively model application behavior, the system utilizes not only permissions but also provider features that manage inter-application data sharing, with particular examination of dangerous feature combinations. The study employed a specially curated dataset to thoroughly characterize both malicious and benign application behaviors. In testing, MFEMDroid achieved 95.38% accuracy, demonstrating its effectiveness in malware detection (*Gu, Hongyan & Hou, 2024*). While the model attained impressive accuracy, its performance hasn't been validated against established benchmark datasets in the literature. The exclusive reliance on permissions and provider features may cause the model to overlook malware that manifests through API calls, network traffic, or opcode patterns.

Android malware detection studies focusing on static feature analysis, the examination remains limited as only specific predefined features can be analyzed, leaving other potential indicators unchecked. To enable faster analysis of all application data, recent research has begun exploring methods to convert application files into visual representations. The remarkable success of deep learning algorithms in image processing has yielded highly effective results in these studies. However, the developed models still require servers with greater processing power than mobile devices to function effectively. There are also no developments aimed at preventing repetitive operation of the model or ensuring model security. *Yadav et al. (2022)* employed convolutional neural networks for malicious software detection. In their study, a deep learning-based approach for Android malware detection using RGB images is proposed. To perform malicious software detection, the EfficientNet architecture developed in a previous study was utilized. EfficientNet convolutional neural networks have relatively fewer parameters compared to MobileNet and ResNet networks while achieving higher success rates on commonly used transfer learning datasets. The proposed method based on EfficientNetB4 achieved a higher performance in Android malware detection from images with a binary classification accuracy of 95.7% compared to the mentioned models (*Yadav et al., 2022*). Although the proposed method achieved an accuracy of 95.7% for binary classification, all the compared models except MobileNetV2 are high-resource-demanding models. MobileNetV2, on the other hand, achieved a lower accuracy of 85%. *Zhu et al. (2023b)* utilized support vector machines and autoencoders. The study proposes an effective and automated method called MEFDroid for Android malware detection. In the proposed MEFDroid method, three different ensemble learning methods, namely Unsupervised Feature Transformation (ESAES), Hybrid Deep Feature Learning (EDAES), and Feature Fusion (EDAFS), were experimented with to identify the method achieving the highest success rate. Experimental results showed that the ESAES, EDAES, and EDAFS algorithms achieved accuracy rates of 93.93%, 95.06%, and 95.14%, respectively (*Zhu et al., 2023b*). The features used to classify the applications in the dataset employed for training and testing the proposed method have not been explained. Additionally, an analysis of performance differences among the ensemble learning methods has not been conducted. This limits the interpretability and applicability of the approach. In the study, a new malicious software framework called MADRF-CNN was developed using DEX images from Android applications. Since the

header and data sections of DEX files did not provide cost-effective information for malicious software detection, they were removed. The filtered DEX files were transformed into a file consisting of hexadecimal numbers. These hexadecimal characters were converted into pixels to form a pixel matrix. Finally, the pixel matrices were written into a JPEG file to complete the image transformation process. The proposed MADRF framework achieved an accuracy rate of 96.9%, which is better than previous studies (*Zhu et al., 2023c*). Although the results obtained are quite impressive, the diversity and complexity of the dataset created for the study remain unclear. Additionally, the performance of the proposed framework has not been validated on a different dataset, leaving the evaluation of its performance on existing datasets incomplete. *Wang, Yu & Yuan (2024)* converted DEX files, AndroidManifest.xml, and API calls extracted from Android applications into grayscale images, which were further processed using techniques such as Canny edge detection, histogram equalization, and adaptive thresholding to generate RGB images. The CICMalDroid 2020 dataset was used for the Android applications to be transformed into images. For Android malware detection from the generated images, GoogleNet and ResNet models were employed. The best result was achieved using the ResNet model, with an accuracy of 97.25% (*Wang, Yu & Yuan, 2024*). *Tang et al. (2024)* proposed a novel malware classification method based on mixed bytecode images and a deep learning attention mechanism. Fixed-width and height grayscale images and Markov images were generated. These grayscale and Markov images were combined using transition probabilities to form a new texture feature space, which helped emphasize the distinguishing features of malware samples. The Drebin and CICMalDroid 2020 datasets were jointly used for training and testing the model. When a convolutional attention mechanism was integrated into the ResNet architecture, the highest accuracy of 98.67% was achieved (*Tang et al., 2024*). Although both *Wang, Yu & Yuan (2024)* and *Tang et al. (2024)* obtained promising results using image-based approaches, the ResNet model employed in both studies is not suitable for deployment on mobile devices.

In the studies examined in the literature, it has been observed that image-based approaches enable the examination of a broader feature set of Android applications. Therefore, this study performs Android malware detection using images. Dalvik Executable (DEX) files have been converted into images to develop an Android malware detection model capable of running on mobile devices. Existing studies demonstrate that approaches using lightweight convolutional neural networks for mobile devices remain insufficient. Furthermore, in no previous study have model predictions and the model itself been stored in secure data storage systems such as blockchain. To address this, our study develops a high-accuracy lightweight convolutional neural network model for mobile devices and stores both the model and its predictions on the blockchain. By enabling the model to run directly on mobile devices, the need for centralized servers is eliminated. The model operates distributively across all devices while being secured *via* blockchain. Consequently, any user can make predictions using the model, which is sufficient for the proposed system to classify Android applications. Storing both the model and its predictions on the blockchain prevents redundant model executions, ensures model

security, enables efficient resource utilization, and prevents potential cyber attacks targeting centralized server resources.

## MATERIALS AND METHODS

Converting Android application DEX files into JPEG images is advantageous for effectively utilizing CNNs in malware detection. Therefore, in the dataset used for malicious software detection, DEX files of Android applications were converted into JPEG images for the training and testing of the CNN model. EfficientNetB0, MobileNetV2, and customized convolutional neural networks CNN capable of running on mobile devices were preferred for training. After training and testing each model separately, a new model was developed by combining the three models using ensemble learning methods. The operations conducted for developing the final model are illustrated in Fig. 1.

As seen in Fig. 1, a mobile application has been developed to utilize the developed ensemble model. The functionality of the developed mobile application is depicted in Fig. 2.

As depicted in Fig. 2, for the classification of Android applications, the first step involves the user uploading the application's APK file to the application. Subsequently, the hash information of the APK file is obtained to check whether the application has been previously analyzed by the model. If the application has been previously analyzed, the user is shown the predicted model result from the blockchain without running the model again. For this purpose, a smart contract has been developed to enable interaction with the blockchain. The developed smart contract includes functions for registering, querying, updating, and deleting records. The register function records the applications analyzed by the malicious software model on the blockchain. Applications are stored in the blockchain with hash, name, SDK, size, and model information. The query function allows querying of the applications analyzed by the malicious software model from the blockchain. Since records cannot be modified in the blockchain, the update function enables the use of the latest record instead of the previous one by creating a new record. Since records cannot be deleted from the blockchain, the delete function allows the data to be treated as if it were not registered in the network during programming. If the application is not registered in the blockchain, the APK file is disassembled, and the DEX file is converted into an image for analysis by the malicious software detection model. After the analysis is completed, the prediction information made by the model is recorded in the blockchain. If the same application is queried again, the prediction information made by the model previously will be used.

The Android malicious software detection model was trained and tested using applications from the CICMalDroid2020 dataset (*Mahdavifar et al., 2020*). The dataset comprises applications from five different categories: adware, banking malware, SMS malware, riskware, and benign software. The choice of the CICMalDroid 2020 dataset is really suitable for the goals of this study, and there are a few key reasons for preferring this dataset. First off, the CICMalDroid 2020 dataset stands out because it has a large number of samples and a high level of diversity. This allows the model to effectively learn and detect different types of malware. The dataset has been meticulously classified using static,

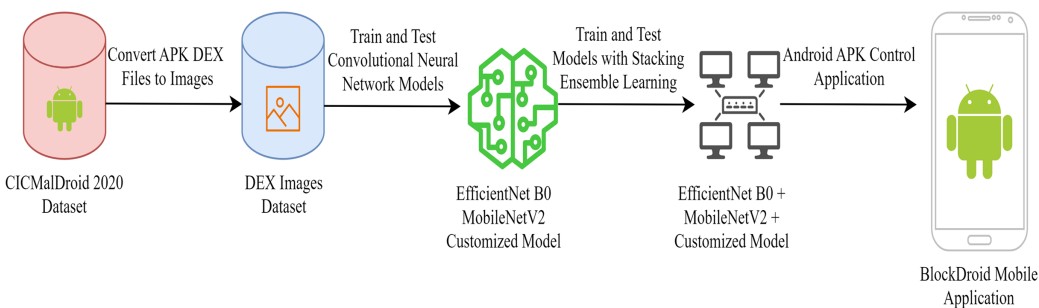

**Figure 1  Android malware detection model training process.**

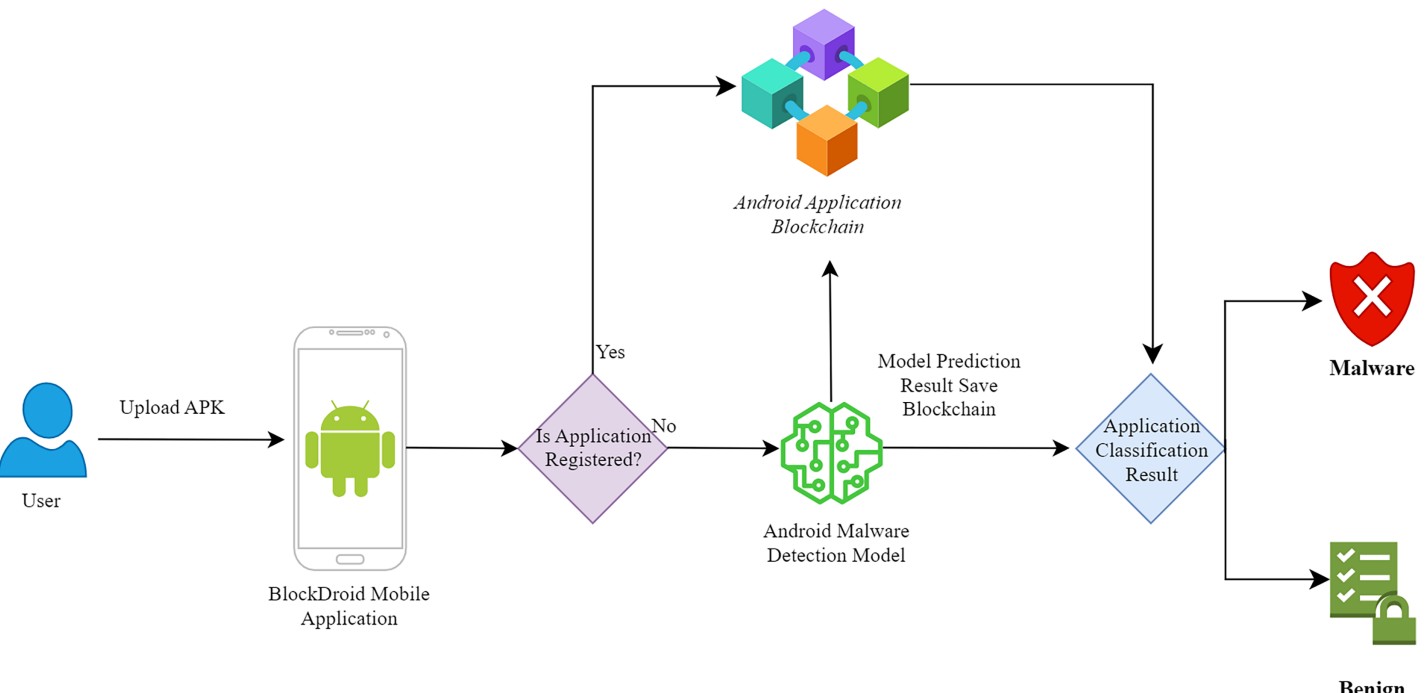

**Figure 2  Android malware detection mobile application flow diagram.**

dynamic, and network traffic analyses during its preparation. Throughout this process, samples from four distinct malware categories—SMS, adware, banking, and riskware—have been collected. This variety aids the model in differentiating between malware applications across various categories. Otherwise, a malware application from an unknown category could have been inaccurately classified. Furthermore, to prevent the model from exhibiting biased decisions towards any specific malware category, efforts have been made to maintain a balanced number of applications across all categories. Likewise, to ensure that the model does not lean towards either malware or benign categories, a balance has been established between the total counts of malicious and benign applications. This approach has significantly enhanced the transparency and reproducibility of the research.

The elucidation of these details plays a crucial role in bolstering the study's reliability and validity. A total of 13,077 applications were analyzed and classified out of 17,341 Android applications in the dataset. Hence, there are a total of 13,077 available applications in the dataset. Among these applications, 1,795 are benign. To balance the dataset, an equal number of 1,795 applications were selected from each of the four categories of malware (449-SMS, 449-adware, 448-banking, and 449-riskware). The DEX files of a total of 3,590 applications in this dataset were converted into images. Among these applications, 1,795 are malicious, and 1,795 are benign applications. The dataset, which comprises APK files, was preprocessed by converting the DEX files within the APKs into JPEG format images. Before being fed into the models for training, these images were standardized to a size of 224 × 224 pixels. Additionally, no data preprocessing has been applied in the conducted study. Eighty percent of the dataset was used for training, while the remaining 20% was used for testing purposes. The split ratio of 0.8 (80%/20% training/test data), known to produce more accurate results in model evaluation (*Rácz, Bajusz & Héberger, 2021*), has been taken into account. This ratio typically provides a robust balance between training model complexity and evaluating performance.

LeCun introduced CNNs in 1989 specifically for image recognition tasks, and since then, CNNs have found extensive applications across diverse domains including image analysis, speech recognition, natural language processing, object or face recognition, and disease detection. Renowned for their efficacy in tasks like image recognition and classification, a typical CNN comprises five layers: input, convolutional, pooling (or subsampling), fully connected, and dropout layers (*Khan et al., 2020*). The input layer, serving as the initial layer, accepts raw data into the CNN. The size of data within this layer significantly impacts model performance. Opting for a larger input image size can elevate memory demands, training time, and test time per image, thus enhancing network performance. Conversely, opting for a smaller input image size reduces memory requirements and training time but may compromise network depth and performance. Hence, selecting an appropriate input image size is pivotal for image analysis, considering both network depth and computational cost (*Şafak et al., 2022*). The primary function of the convolutional layer is feature extraction from the input image. This layer learns image features using small portions of input data, preserving spatial relationships between pixels. The input image undergoes convolution using learnable filters or kernel sets, with each filter generating a feature map in the output image. These feature maps proceed to the subsequent convolutional layer. This convolution process entails applying specific filters across the entire image, making filters indispensable components of the layered architecture. Filters can vary in size, such as 2 × 2, 3 × 3, and 5 × 5, and apply convolution to the input from the preceding layer to produce the output of the convolutional layer, resulting in an activation map (Feature map) where specific features are detected for each filter. During training with a CNN, the coefficients of these filters evolve with each learning iteration on the training set, enabling the network to discern crucial data regions for feature determination (*Li et al., 2022*). The pooling layer primarily aims to reduce computational costs by downsizing the feature map. This involves reducing connections between layers, allowing each feature map to operate independently. Various pooling

operations exist, including maximum pooling, average pooling, and total pooling. Typically bridging the convolutional layer and the fully connected layer, the pooling layer facilitates computational efficiency (*Elmas, 2021*). The fully connected layer, positioned before the output layer, comprises neurons with associated weights and biases, establishing connections between neurons across different layers. Flattening the input image from previous layers, it feeds into the fully connected layer (*Khan et al., 2020*), where mathematical function operations usually occur, initiating the classification process. To prevent overfitting, a dropout layer is employed during the training process, randomly dropping a percentage of neurons in the neural network, thereby reducing model size. Activation functions play a crucial role in CNN models, enabling them to learn and predict complex relationships between network variables, adding non-linearity to the network. Commonly used activation functions include ReLU, Softmax, tanH, and Sigmoid functions (*Şafak & Barişçi, 2023*). When selecting models for training, lightweight models such as EfficientNetB0 and MobileNetV2, which have low parameter counts and can run on mobile devices, were preferred. The custom-developed model was also designed to ensure that the number of parameters would not prevent it from running on mobile devices. Lightweight CNN models used to ensure the feasibility of running the model on mobile devices. Achieving high performance with lightweight CNN models requires fine-tuning hyperparameters. However, improvements through hyperparameter tuning can only enhance model performance to a limited extent. In the experiments conducted, the feasibility of the developed models was validated using the developed mobile application. The choice of blockchain technology was driven by the need for a permissioned and high-performance infrastructure. We selected Hyperledger Fabric due to its superior performance compared to other blockchain platforms. However, its setup and integration can be challenging. We encountered issues related to version compatibility and smart contract code during development.

The MobileNetV2 is a convolutional neural network designed to efficiently operate on mobile and embedded devices while aiming for better performance. It consists of 53 layers and 3.4 million parameters. The MobileNetV2 architecture includes a first fully convolutional layer with 32 filters, followed by 19 residual bottleneck layers. Additional residual connections and bottleneck layers have been added to the basic MobileNet architecture. The bottleneck residual block is inserted between the layers of the network. Instead of depthwise separable convolution used in the MobileNet architecture, the bottleneck residual block in MobileNetV2 has been developed. The bottleneck residual block enables the network to compute activations more efficiently and preserve more information after activation. Unlike the pointwise convolutions in the MobileNet architecture, the bottleneck residual blocks in MobileNetV2 reduce the number of channels. Moreover, because the bottleneck residual layers have a linear structure, they prevent excessive information loss from non-linear layers (*Sandler et al., 2018*). The general architecture of MobileNetV2 is outlined in Fig. 3.

EfficientNet introduces a novel scaling approach employing compound coefficients within the architecture of convolutional neural networks. Unlike traditional scaling methods that arbitrarily scale network dimensions such as width, depth, and resolution,

| Input | Operator | $t$ | $c$ | $n$ | $s$ |
|---|---|---|---|---|---|
| $224^2 \times 3$ | conv2d | - | 32 | 1 | 2 |
| $112^2 \times 32$ | bottleneck | 1 | 16 | 1 | 1 |
| $112^2 \times 16$ | bottleneck | 6 | 24 | 2 | 2 |
| $56^2 \times 24$ | bottleneck | 6 | 32 | 3 | 2 |
| $28^2 \times 32$ | bottleneck | 6 | 64 | 4 | 2 |
| $14^2 \times 64$ | bottleneck | 6 | 96 | 3 | 1 |
| $14^2 \times 96$ | bottleneck | 6 | 160 | 3 | 2 |
| $7^2 \times 160$ | bottleneck | 6 | 320 | 1 | 1 |
| $7^2 \times 320$ | conv2d 1x1 | - | 1280 | 1 | 1 |
| $7^2 \times 1280$ | avgpool 7x7 | - | - | 1 | - |
| $1 \times 1 \times 1280$ | conv2d 1x1 | - | k | - | |

**Figure 3 General architecture of MobileNetV2.**

the compound scaling method uniformly scales each network dimension using a fixed scaling coefficient. This approach has demonstrated enhanced model accuracy and efficiency compared to conventional scaling methods. With compound scaling, it is established that larger input images necessitate more channels to detect additional layers and finer details within the larger image. The EfficientNet architecture predominantly employs mobile inverted bottleneck convolution (MBConv). EfficientNetB0 represents an adapted version of the EfficientNet network optimized for mobile and embedded devices, comprising 5.3 million parameters. Alongside squeeze-and-excitation blocks, EfficientNetB0 integrates mobile inverted bottleneck residual blocks akin to those utilized in the MobileNetV2 network (*Tan & Le, 2019*). The architecture of EfficientNetB0 is outlined in Fig. 4.

- The following configuration was applied for training MobileNetV2 and EfficientNetB. MobileNetV2 has enhanced its effectiveness by adding a flatten layer and a dense layer with a sigmoid activation function. The flatten layer transforms feature maps into a one-dimensional vector, making it suitable for the dense layer. The dense layer computes weight values, and the sigmoid function facilitates classification by adjusting the model's output to the [0, 1] range, classifying images as malware or benign.
- Similarly, for EfficientNetB0, the same enhancements are applied: a flatten layer and a dense layer with a sigmoid activation function. This setup ensures consistency in processing and classification across different models.

For Android malware detection, a medium-density model consisting of a total of 10 layers, including three convolutional layers, three max-pooling layers, one flattening layer, two fully connected layers, and one dropout layer, was obtained through experiments. The

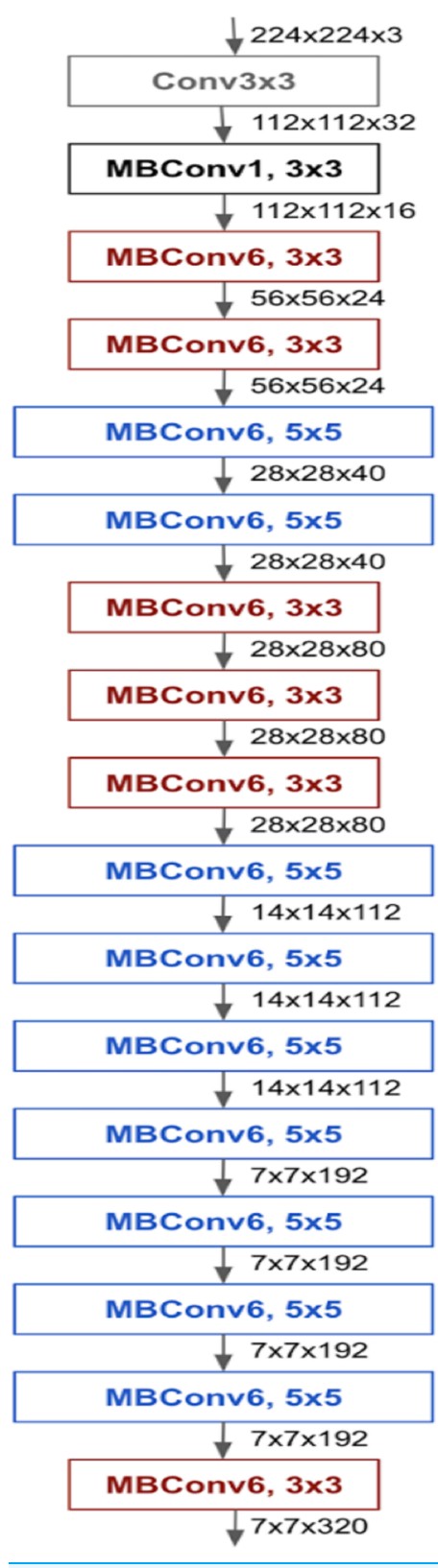

**Figure 4 Architecture of EfficientNetB0.**

model was designed to keep the number of parameters minimum while maximizing the success rate. The architecture of the model is shown in Fig. 5.

Ensemble learning is a machine learning technique that harnesses the amalgamation of multiple models to enhance performance. Among the most prominent ensemble learning methods are Bagging, Boosting, and Stacking. Bagging involves training multiple models on various samples of the same training dataset. Boosting, on the other hand, entails multiple models making sequential predictions for each example during training, with the weighted average of predictions from all models being computed after each model's prediction is utilized by the subsequent model (*Wen & Hughes, 2020*). Stacking, the focus of this study, involves training multiple models separately on the same dataset, and during the prediction phase, the predictions of these models are amalgamated. The efficacy of ensemble learning lies in the diversity of operation among individual machine learning models. While one model may excel on certain data, others may exhibit less effectiveness. By amalgamating diverse models, ensemble learning mitigates the weaknesses of individual models (*Cui et al., 2021*). In this study stacking ensemble learning method using a logistic regression algorithm as the meta-model. The predictions from the ensemble models are input into logistic regression, which learns the final classification. Logistic regression is effective for binary classification tasks, leveraging mathematical techniques to model relationships between a dependent variable and a set of independent variables. It is particularly chosen for its efficiency in terms of computational resources like memory and processing power.

TensorFlow, an open-source software library for machine learning, was developed by Google and unveiled in 2015. It functions as a symbolic mathematics library grounded in data flow and differentiable programming principles. With its second version, launched in 2017, TensorFlow introduced support for CPU and GPU processing. Versatile in application, TensorFlow operates seamlessly across various computing platforms, including 64-bit Linux, macOS, Windows, Android, and iOS. Its adaptable architecture facilitates the efficient distribution of computation across diverse platforms, spanning from desktops to server clusters and from mobile to edge devices (*Tensorflow, 2024*). The term "TensorFlow" derives from the tensor operations performed by neural networks, where tensors represent multi-dimensional data arrays. Tensors generalize vectors and matrices to potentially higher dimensions. Data arrays of varying dimensions and sequences fed into neural networks are termed tensors. Given the intricacies of deep learning, especially during training, TensorFlow's computations are conceptualized as data flow graphs incorporating state information (*Janardhanan, 2020*). TensorFlow consists of three main components: the TensorFlow Framework, which provides functions for defining models and developing data models with a user-friendly Python interface; TensorBoard, a tool for analyzing, visualizing, and debugging TensorFlow graphs; and TensorFlow Serving, a flexible, high-performance service system for deploying trained deep learning models in production. Predominantly written in C++ with a Python interface, TensorFlow is widely used in both academic research and industrial applications, making it the preferred library for this study (*Şeker, Diri & Balık, 2017*).

Blockchain is a decentralized distributed ledger technology. In a blockchain, data is added linearly in blocks after being validated. Each block consists of two main parts: a

```
Layer (type)              Output Shape          Param #
=================================================================
conv2d (Conv2D)           (None, 148, 148, 32)  896

max_pooling2d (MaxPooling2 (None, 74, 74, 32)   0
D)

conv2d_1 (Conv2D)         (None, 72, 72, 64)    18496

max_pooling2d_1 (MaxPoolin (None, 36, 36, 64)   0
g2D)

conv2d_2 (Conv2D)         (None, 34, 34, 128)   73856

max_pooling2d_2 (MaxPoolin (None, 17, 17, 128)  0
g2D)

flatten (Flatten)         (None, 36992)         0

dense (Dense)             (None, 512)           18940416

dropout (Dropout)         (None, 512)           0

dense_1 (Dense)           (None, 1)             513

=================================================================
Total params: 19034177 (72.61 MB)
Trainable params: 19034177 (72.61 MB)
Non-trainable params: 0 (0.00 Byte)
```

**Figure 5  Customized model architecture.**

header and data. Every block contains the hash of the previous block, ensuring the integrity of the chain. Validators add legitimate transactions to the blockchain network using consensus algorithms. Validations are performed in the order of transactions, but adding data to the network depends on the completion time of validation. A copy of the blockchain network is maintained by all relevant participants. Data in the blockchain cannot be altered by any participant. Blockchain has fundamental properties, such as immutability, privacy, security, trustlessness, and decentralization (*Şafak, Arslan & Gözütok, 2020*). Immutability ensures that data added to the blockchain network cannot be altered. Immutability is achieved by applying a hash function to the blocks within the blockchain, timestamping the blocks, and chaining them together. Security ensures that data within the blockchain cannot be viewed by unauthorized individuals. Data within the blockchain network is stored encrypted, and no participant can modify blockchain records. Trustlessness enables participants to transact with each other without the need for a central authority. Transactions within the blockchain network are executed through consensus algorithms. Privacy ensures that the party performing a transaction in the blockchain network cannot be identified. Blockchain transactions are stored anonymously, thus preserving the privacy of users. Distributed Architecture refers to the blockchain network's copy being maintained by multiple participants. In public blockchain networks, the database is present in all participants, while in permissioned blockchain networks, the

database is present only in relevant participants (*Cagigas et al., 2021*). With the increasing adoption of blockchain technology, smart contracts have emerged. Smart contracts enable transactions to be executed autonomously, quickly, and securely within the blockchain network. Smart contracts allow blockchain technology to be utilized in various sectors and complex transactions. By minimizing manual operations, smart contracts reduce errors and lower costs. In blockchain technology, data is stored across validating participants instead of centralized servers,which reduces the burden on centralized servers. Participants can transact with each other without intermediaries. While traditional blockchain applications are typically designed to be open to everyone, there are also private blockchain approaches where participation is permission-based (*Şafak, Mendi & Erol, 2019*).

Hyperledger Fabric is a permissioned private blockchain platform used in various enterprise-level scenarios. It is developed and supported by the Linux community. Thanks to its modular architecture, it finds applications in diverse fields, such as banking, finance, insurance, healthcare, supply chain, auctions, defense, and more. It supports smart contracts in Java, Go, and Node.js programming languages. With the addition of the Byzantine Fault Tolerant (BFT) protocol in Hyperledger Fabric v3, the blockchain network is strengthened against potential attacks in case the orderer node is compromised. Attacks that can occur when the orderer node is compromised include not sending transactions to the network, sending desired blocks to desired nodes, and sending different blocks to different nodes (*Antwi et al., 2021*). Currently, Hyperledger Fabric supports the Raft consensus protocol. Raft is a consensus algorithm where transactions are confirmed by the leader node selected among the nodes and distributed to other nodes. Although Hyperledger Fabric 2.5 does not support Byzantine fault tolerance, it supports crash fault tolerance (CFT), which prevents possible repeated transactions before the network is updated. Significant improvements have been made in terms of performance in Hyperledger Fabric 2.5. In a test conducted with two peers and one orderer node, it reached a TPS value of 2,946 in simple asset creation. Hyperledger Fabric v3 roadmap includes improvements for performance and protection against quantum attacks (*Al-Sumaidaee et al., 2023*). Due to these features distinguishing it from other blockchain infrastructures, Hyperledger Fabric was used in the conducted study. Hyperledger Fabric version 2.5.6 was used. Hyperledger Fabric network was set up with a configuration consisting of two peers and one orderer node across two separate organizations. CouchDB was used as the state database. Hyperledger Fabric blockchain network was deployed with its default network configuration, which includes two organizations and one orderer node. Organization nodes receive requests from users and send them to the orderer node, which then adds transactions to blocks and records them on the blockchain. Transactions were conducted with a user connected to Organization 1 to record the checked applications on the blockchain. A smart contract was developed using the Go language to facilitate these transactions, including functions such as Save, Query, Update, Delete, and Check. Applications identified as malicious are recorded using the Save function with details like hash, name, SDK, size, and model information. In the Android malware detection application, before running the deep learning model, it checks whether the application is already recorded on the blockchain using the Check function with the application's hash information.

The performance of CNNs is significantly influenced by several hyperparameters, including the number of filters, filter size, number of layers, dropout rate, and learning rate. The values in Table 1 indicate that the proposed model achieved maximum success rates. The epoch value represents how many times the dataset was trained on the proposed CNN. In this study, the dataset reached its maximum value after being trained 250 times on the proposed CNN, achieved using early stopping in Python. Early stopping halts training when improvements in model performance cease. The learning rate, which determines the rate at which learned weights are updated, was set to 0.001. To calculate the loss between the model's predictions and actual values, the binary cross-entropy function was used. This loss function is proven effective in preventing errors caused by noisy data in large datasets (*Zhang & Sabuncu, 2017*). The Adam optimization algorithm was chosen for its computational efficiency, low memory requirements, and suitability for problems with large datasets, allowing the learning rate to be updated based on different parameters (*Kingma & Ba, 2017*). The hyperparameters listed in Table 1 are used for training the models.

Evaluation of model performance is vital to understand how the model performs on real-world data. Therefore, the following evaluation metrics are used in the study.

Accuracy represents the ratio of correct predictions to the total number of observations. This metric is essential for assessing the overall success of our malware detection model. It indicates the proportion of correct predictions (both true positives and true negatives) out of all predictions made. Although useful, accuracy alone can be misleading in imbalanced datasets, which is why additional metrics are considered.

$$\text{Accuracy} = (\text{Number of correct predictions})/(\text{Total number of predictions}). \tag{1}$$

Recall, represents the ratio of true positives to the total number of positive examples, signifies the proportion of true positives among all positives. It holds significance in scenarios where false negatives carry significant weight. Also known as sensitivity, recall reflects the model's ability to detect all actual malware applications. A high recall ensures that the model does not miss potential threats, which is vital in a security context to minimize undetected malware.

$$\text{Recall} = (\text{True Positive})/(\text{True Positive} + \text{False Negative}). \tag{2}$$

Precision, alternatively referred to as positive predictive value, signifies the ratio of true positives to the total predicted positives. It denotes the proportion of true positives among all positive predictions. Precision holds importance in scenarios where false positives are of utmost concern. Precision measures the ratio of correctly identified malware applications to all applications predicted as malware. A high precision indicates a low false positive rate, meaning the system is reliable and unlikely to trigger false alarms, which is crucial for maintaining user trust and system efficiency.

$$\text{Precision} = (\text{True Positive})/(\text{True Positive} + \text{False Positive}). \tag{3}$$

F1 score, defined as the harmonic mean of precision and recall, equally emphasizes both precision and recall, rendering it beneficial in instances of imbalanced class distributions.

**Table 1  Hyperparameters used in training.**

| Hyperparameter | Value |
| --- | --- |
| Epoch | 250 |
| Activation function | ReLU |
| Learning rate | 0.001 |
| Loss_function | Binary_crossentropy |
| Optimizer | Adam |
| Threshold | 0.5 |

F1 score is a harmonic mean of precision and recall, providing a balance between them. It is particularly valuable when the class distribution is uneven, as it helps to ensure that both false positives and false negatives are minimized. A high F1 score indicates that the model effectively identifies malware without compromising on either precision or recall.

$$F1 \ score = 2 \times (Precision \times Recall)/(Precision + Recall). \tag{4}$$

## RESULTS AND DISCUSSION

In this study, Python 3.8 and the TensorFlow library were used. The experiments were conducted using a computer equipped with an Intel(R) Core(TM) i7-12700H CPU, NVIDIA® GeForce® RTX4060 GPU, and 16GB RAM hardware, running the Ubuntu 22.04 operating system. For the training process, 2,872 images were used, while 718 images were used for the testing process. Initially, lightweight models capable of running on mobile devices, namely MobileNetV2 and EfficientNetB0, were trained and tested. Subsequently, a customized model was developed to achieve higher accuracy. The developed customized model consists of 10 layers, including three convolutional layers, three maximum pooling layers, one flattening layer, one dropout layer, and two fully connected layers. While developing the model for high performance on mobile devices, attention was paid to keeping the number of parameters to a minimum. Finally, the developed MobileNetV2, EfficientNetB0, and customized models were used together with the stacking ensemble learning method. The performance metrics obtained from the experiments are presented in Table 2.

As seen in Table 2, the highest accuracy rate of 93.14% was achieved using the customized model among the models tested. Subsequently, by training the MobileNetV2, EfficientNetB0, and customized model together with the stacking ensemble learning method, the malware detection model with the highest accuracy rate of 96.17% was obtained. To further validate the model's performance, we employed k-fold cross-validation, which involves dividing the dataset into 'k' equal parts. This technique allows the model to be trained and tested across multiple iterations, thereby improving its ability to generalize to unseen data. Although there is no definitive rule for selecting the value of 'k', values of 5 and 10 are commonly used. The model's performance was measured using k = 5 and k = 10, with results indicating consistent performance across both settings, as shown in Table 3.

**Table 2 Performance metrics of malware detection models.**

|  | Accuracy (%) | Precision (%) | Recall (%) | F1 score (%) |
| --- | --- | --- | --- | --- |
| MobileNetV2 | 80.07 | 75.45 | 79.57 | 77.45 |
| EfficientNetB0 | 84.82 | 76.13 | 82.92 | 79.38 |
| Customized model | 93.14 | 84.76 | 90.70 | 87.62 |
| MobileNetV2 + EfficientNetB0 + Customized model | 96.17 | 92.50 | 95.43 | 93.94 |

The performance metrics seen in Table 3 demonstrate robust precision, recall, and F1 score for the MobileNetV2 + EfficientNetB0 + Customized Model stacking ensemble approach, affirming the model's effectiveness in malware detection. The comparison of the study with the one conducted using the CICMalDroid2020 dataset can be seen in Table 4.

As seen in Table 4, the proposed model achieved a higher accuracy rate compared to previous studies. The comparison of the proposed model with the latest state-of-the-art models from 2023 and 2024 is presented in Table 5.

As seen in Table 5, the proposed model has achieved competitive performance compared to the latest models. The comparison of Android malware detection studies using images with the proposed model is shown in Fig. 6.

As seen in Fig. 6 the proposed model has achieved a higher accuracy compared to the using image studies. Following the development of the malware detection model, an application was developed using Flask to enable the inspection of Android applications and the recording of model predictions to the blockchain. The Android application control screen is shown in Fig. 7.

As seen in Fig. 7, the APK file of the Android application can be uploaded through the application to check whether the application is secure or malicious. When the Install Application button is used to install the application, the hash information is first obtained, and it is queried from the blockchain network whether it has been previously checked by the model. If the application has not been previously checked by the model, its DEX file is parsed and converted into an image for analysis by the model. The analysis result of the application being checked for the first time is shown in Fig. 8.

In Fig. 8, the APK file of the WhatsApp application has been uploaded as an example. Upon uploading, the model has predicted it as benign (safe). Since the application is being checked for the first time, it is observed that it is not recorded in the blockchain. The model prediction has been recorded in the blockchain, and the application has been saved. Additionally, information such as the application name, SDK version, and file size is displayed. The result of rechecking the WhatsApp application in the last attempt is shown in Fig. 9.

As seen in Fig. 9, since the same application was reanalyzed, the application was not re-run as it was already registered in the blockchain, and the analysis result was retrieved from the blockchain network. This prevented computational resources from being occupied for the same application.

**Table 3  The performance metrics of the proposed model obtained through k-fold cross-validation.**

|         | Accuracy (%) | Precision (%) | Recall (%) | F1 score (%) |
|---------|--------------|---------------|------------|--------------|
| 80–20%  | 96.17        | 92.50         | 95.43      | 93.94        |
| K = 5   | 97.38        | 95.64         | 96.55      | 96.09        |
| K = 10  | 96.85        | 95.12         | 96.26      | 95.68        |

**Table 4  Comparison of the proposed model with the previous study.**

|                                              | Accuracy (%) | Precision (%) | Recall (%) | F1 score (%) |
|----------------------------------------------|--------------|---------------|------------|--------------|
| Proposed model                               | 97.38        | 95.64         | 96.55      | 96.09        |
| *Atacak, Kılıç & Doğru (2022)*               | 92           | 92.15         | 92         | 92.01        |
| *Sönmez, Salman & Dener (2021)*              | 91.4         | 91.6          | 91.1       | 91.4         |
| *Padmavathi, Shanmugapriya & Roshni (2022)*  | 88           | –             | –          | –            |
| *Kou et al. (2023)*                          | –            | –             | –          | 85.49        |
| *Wang, Yu & Yuan (2024)*                     | 97.25        | 95.66         | 95.77      | 95.71        |

**Table 5  The comparison of the proposed model with the latest state-of-the-art models from 2023.**

|                                      | Accuracy (%) | Precision (%) | Recall (%) | F1 score (%) |
|--------------------------------------|--------------|---------------|------------|--------------|
| Proposed model                       | 97.38        | 95.64         | 96.55      | 96.09        |
| *Zhu et al. (2023b)*                 | 95.14        | 97.29         | 96.94      | 97.12        |
| *Zhu et al. (2023c)*                 | 96.9         | 97.1          | 98.9       | 95.9         |
| *Faghihi, Zulkernine & Ding (2023)*  | 98.8         | 98.9          | 98         | 99.3         |
| *Huang et al. (2023)*                | 92.50        | 93.10         | 91.80      | 92.45        |
| *Zhu et al. (2023a)*                 | 96.48        | 97.62         | 97.05      | 96.07        |
| *Gu, Hongyan & Hou (2024)*           | 95.38        | 95.78         | 94.47      | 95.12        |
| *Wang, Yu & Yuan (2024)*             | 97.25        | 95.66         | 95.77      | 95.71        |
| *Tang et al. (2024)*                 | 98.67        | 98.62         | 98.67      | 98.64        |

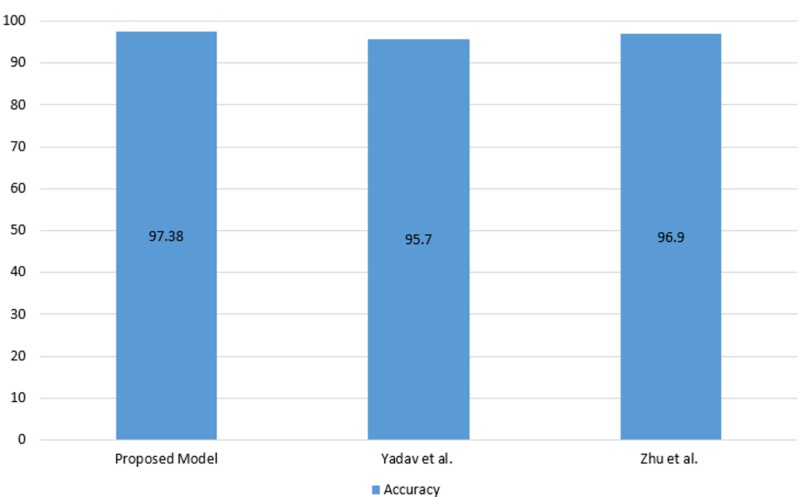

**Figure 6  Comparison of Android malware detection studies using images with the proposed model.**

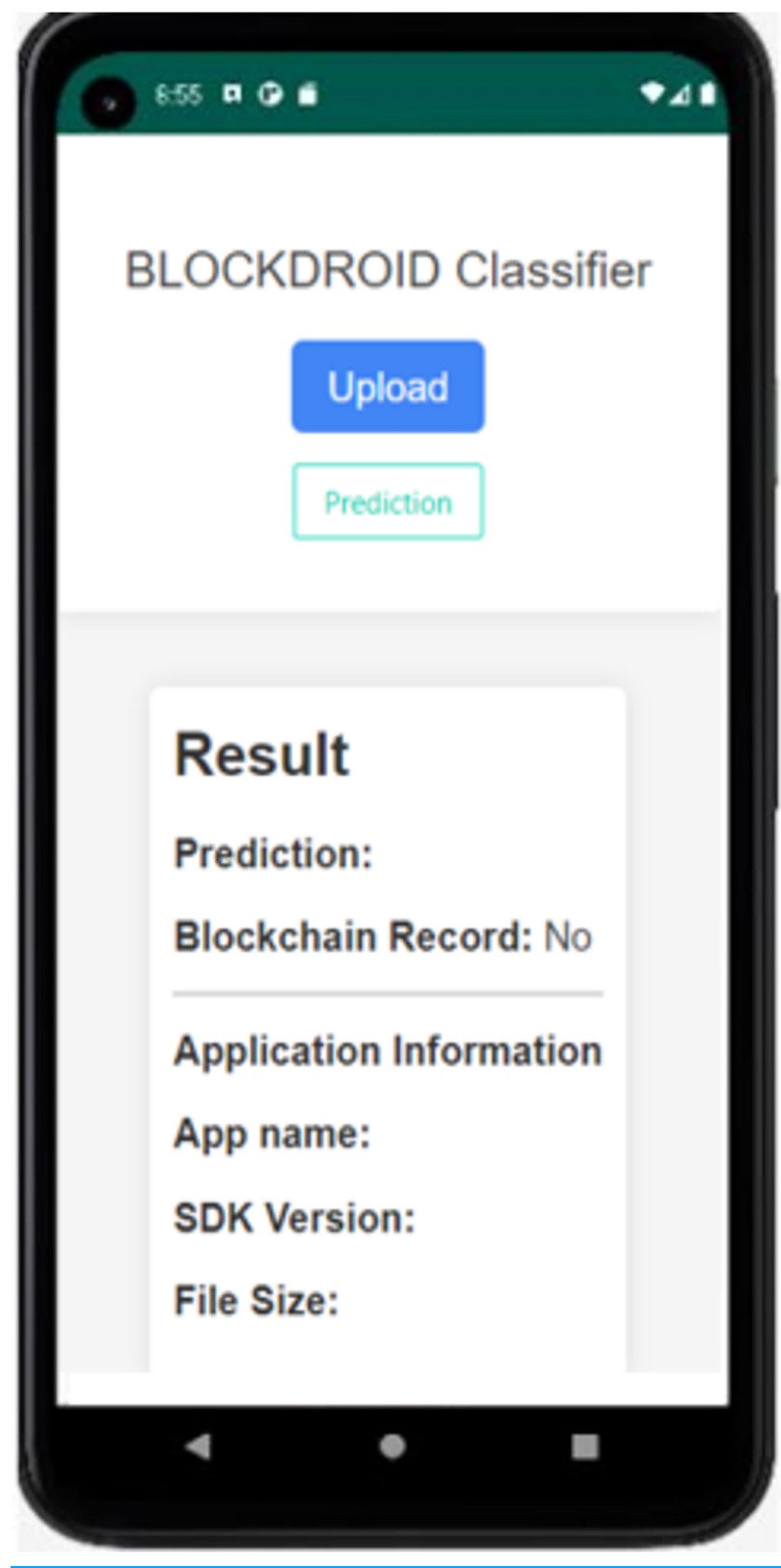

**Figure 7** **BlockDroid mobile application screen.**

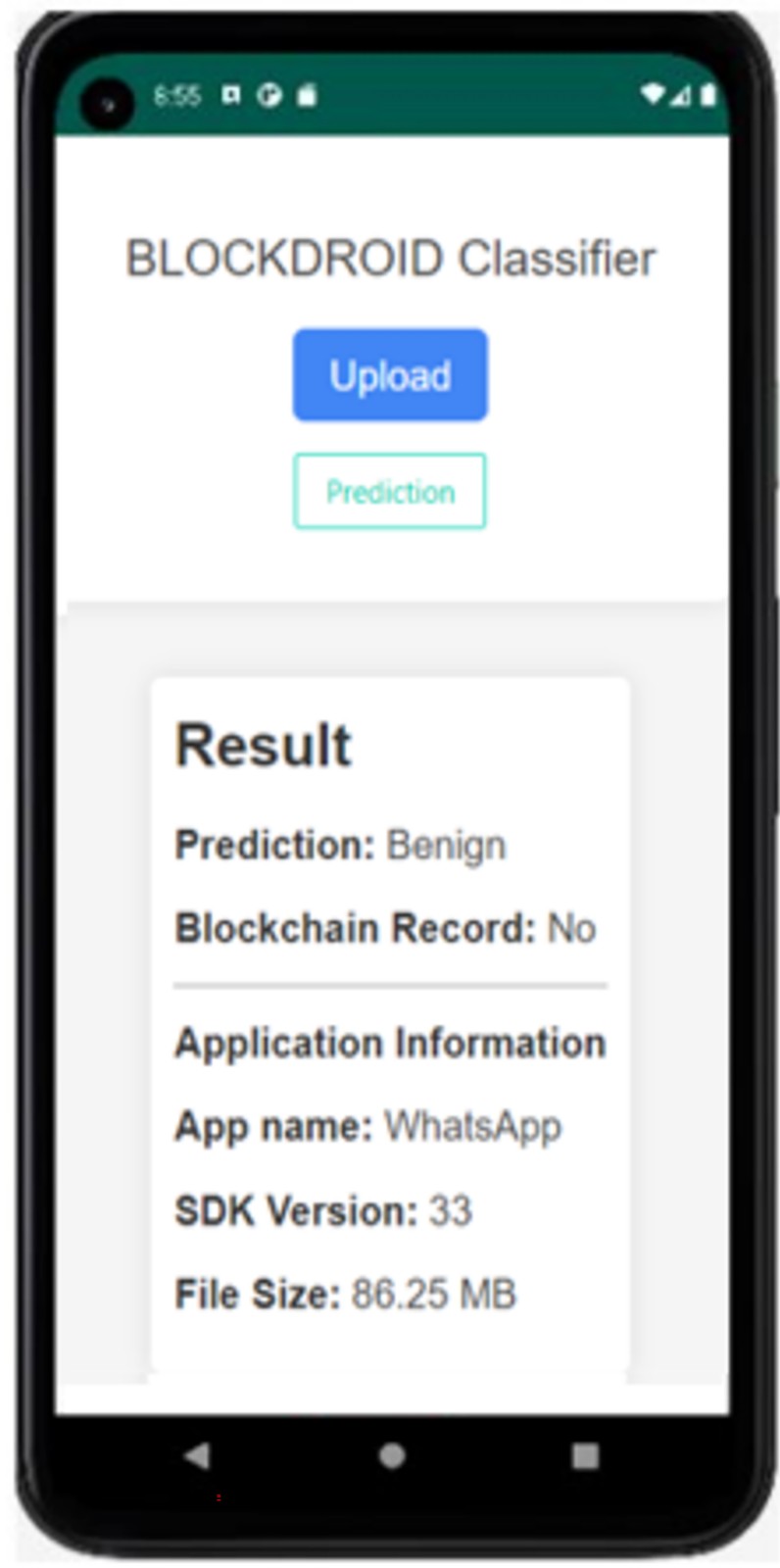

**Figure 8  The result of the first time analyzed application.**

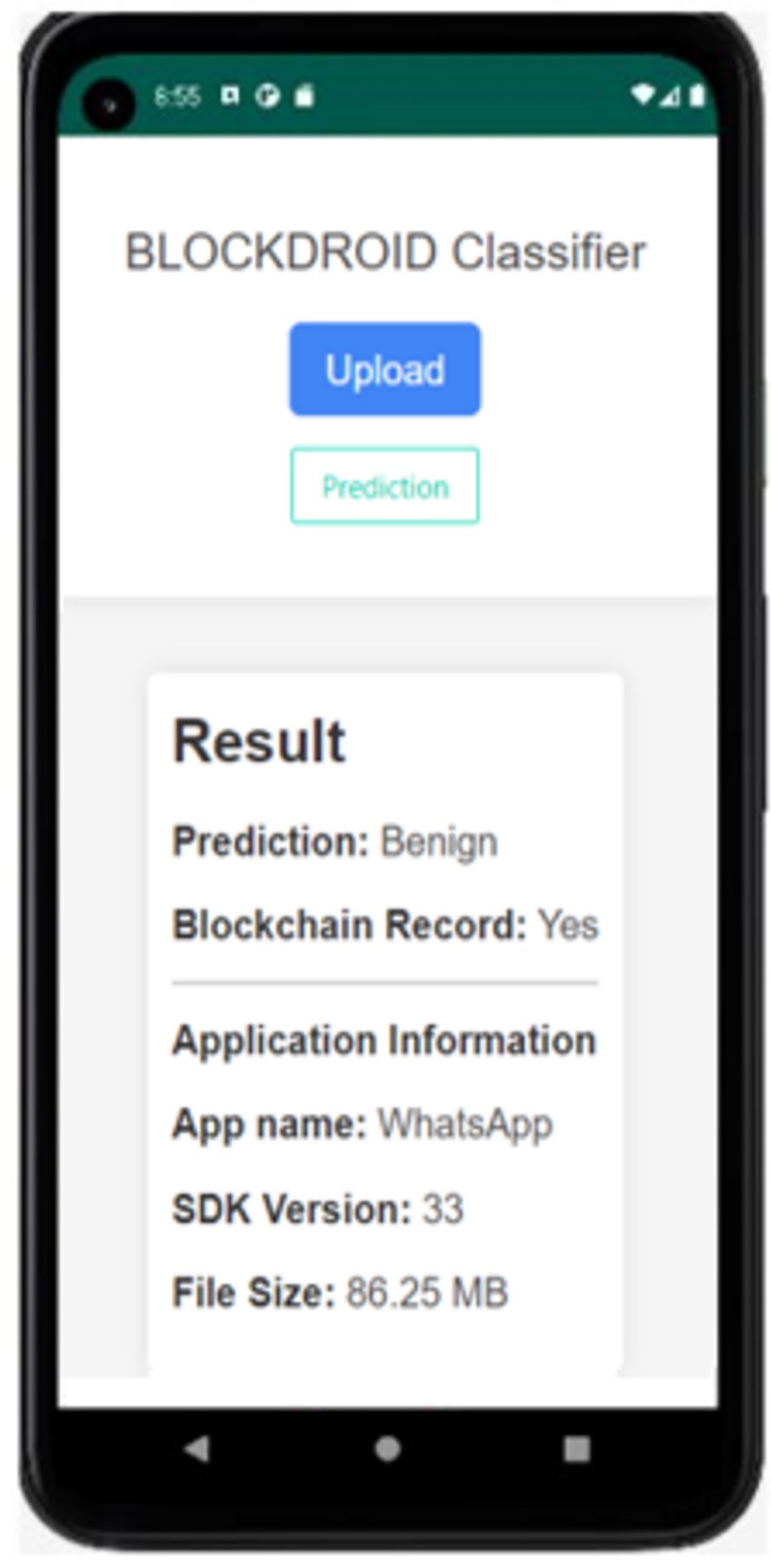

**Figure 9** **The result of the application reanalyzed.** Full-size 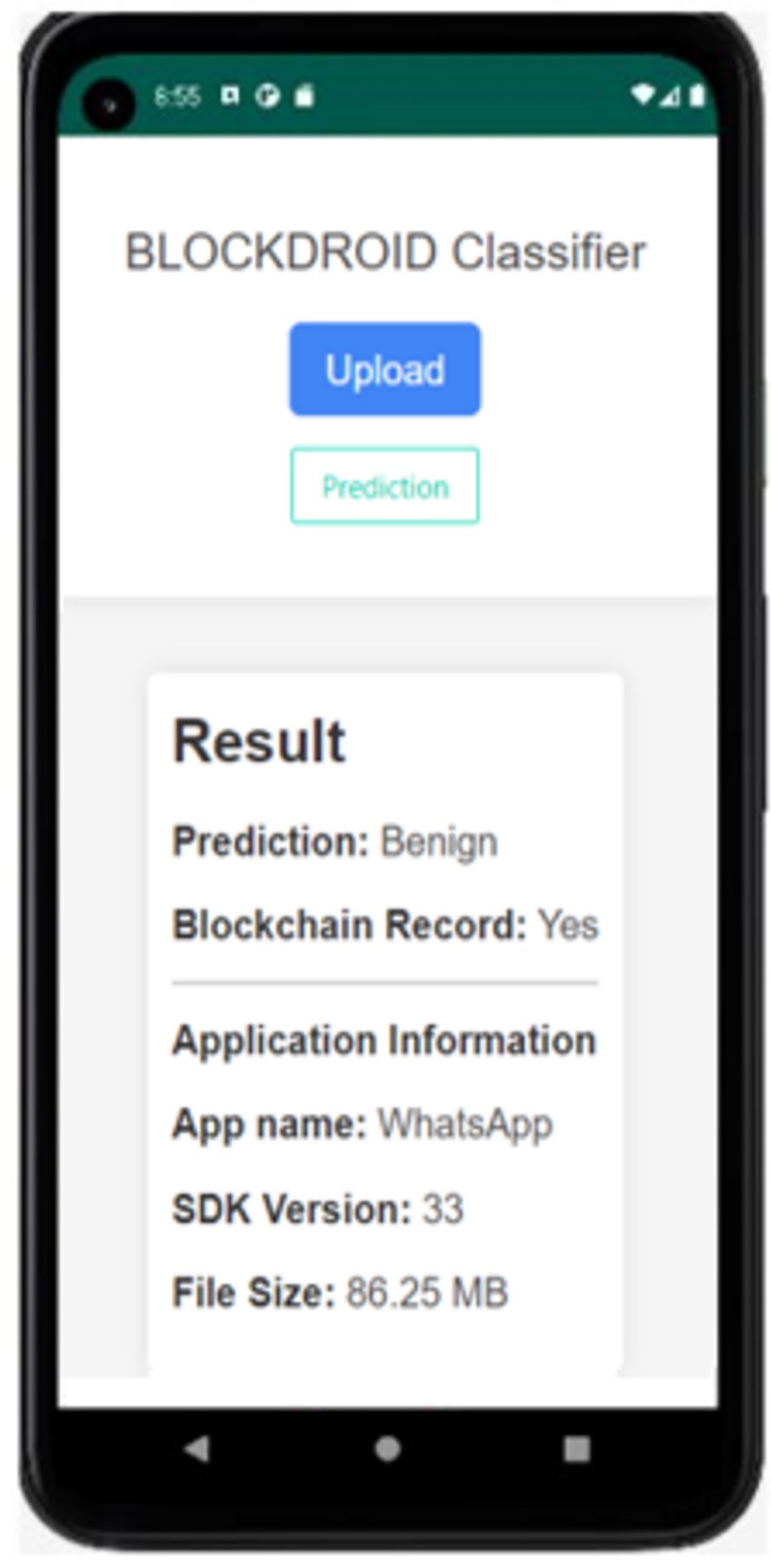 DOI: 10.7717/peerj-cs.2918/fig-9

## CONCLUSIONS

Android, being the most prevalent mobile operating system, is a prime target for attackers, given its widespread adoption on mobile devices. Consequently, research into Android malware detection has become imperative due to the escalating quantity and sophistication of Android malware. In recent investigations, there has been a growing trend towards employing machine learning and deep learning techniques for malware detection. Particularly noteworthy is the increasing success of deep learning across domains such as image recognition/classification, natural language processing, signal processing, and speech processing, which has further bolstered its application in malware detection. Therefore, in this study, deep learning was utilized to detect malware from images. Lightweight models such as MobileNetV2, EfficientNetB0, and a customized model were used to ensure the developed model's compatibility with mobile devices. When the ensemble learning method was applied to these three models together, the highest accuracy rate of 97.38% was achieved. Subsequently, blockchain technology, a secure storage technology that allows the recording of predictions, was employed to prevent the repeated operation of the malicious detection model. Hyperledger Fabric was chosen as the underlying blockchain infrastructure. If an application previously analyzed by the model is rechecked, the analysis result of the application is retrieved from the blockchain without running the model again. This prevents the repetitive operation of the malicious software detection model and ensures efficient resource utilization. This study also serves as an important indicator for the combined use of blockchain and artificial intelligence technologies. Our results are based on static analysis and the conversion of DEX files into images, which are areas open for improvement. Static analysis alone cannot detect real-time behaviors of malicious applications, as some malware exhibits dynamic behaviors that can only be identified during execution. Since the malware detection process from images is a static analysis method, it does not take into account the dynamic behavior of the application during execution. Additionally, the model's predictions may vary depending on the quality of the images produced from the DEX files. The quality of images significantly impacts detection accuracy. The better and more diverse the images in the dataset, the higher the model's ability to make accurate predictions and generalize. In the future, the continuous improvement of the malware detection model can be achieved by entering application analysis results from trusted nodes into the blockchain, allowing the model to learn from the data in the blockchain. Future work could involve dynamic analysis to establish a hybrid framework, combining both static and dynamic analysis to improve detection capabilities. Enhancing the image quality by converting DEX files into higher resolution images could potentially increase the model's accuracy. Additionally, the malware detection model can be enhanced to consider different parameters, such as permissions, APIs, intents, and transaction codes alongside images, when making decisions.

### Funding

The authors received no funding for this work.

### Competing Interests

Emre Şafak is an employee of the HAVELSAN.

### Author Contributions

- Emre Şafak conceived and designed the experiments, performed the experiments, analyzed the data, performed the computation work, prepared figures and/or tables, authored or reviewed drafts of the article, and approved the final draft.
- İbrahim Alper Doğru conceived and designed the experiments, performed the experiments, analyzed the data, prepared figures and/or tables, authored or reviewed drafts of the article, and approved the final draft.
- Necaattin Barışçı conceived and designed the experiments, performed the experiments, analyzed the data, prepared figures and/or tables, authored or reviewed drafts of the article, and approved the final draft.
- İsmail Atacak conceived and designed the experiments, performed the experiments, analyzed the data, prepared figures and/or tables, authored or reviewed drafts of the article, and approved the final draft.

### Data Availability

The Android malware dataset (CICMalDroid 2020) is available at: https://www.unb.ca/cic/datasets/maldroid-2020.html.

The DEX images are available at figshare: Şafak, Emre (2024). Android DEX Images. figshare. Dataset. https://doi.org/10.6084/m9.figshare.26369578.v1.

### Supplemental Information

Supplemental information for this article can be found online at http://dx.doi.org/10.7717/peerj-cs.2918#supplemental-information.

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
