# Peer review of "BlockDroid: detection of Android malware from images using lightweight convolutional neural network models with ensemble learning and blockchain for mobile devices"

_PeerJ Computer Science, doi:10.7717/peerj-cs.2918_

## Round 0.1 · original submission · Major Revisions

Dear Authors,

Your paper has been revised. Given the criticisms of the reviewer(s), your manuscript needs major revisions before being considered for publication in PEERJ Computer Science. Please make changes to your manuscript based on the reviewers' comments before uploading your updated manuscript. In particular, you have to carefully solve the following issues:

1) Your paper needs a more comprehensive explanation of the metrics used, clarifying why each is relevant to malware detection.
2) More details regarding the division of data into training and testing sets, along with any cross-validation techniques employed, must be added to the revised version of your work to enhance the credibility of the reported results.

Thank you for your fine contribution.

·

Basic reporting

Clarity and Language: The paper is written in professional, clear, and unambiguous English, suitable for an international audience. The abstract and introduction sections effectively present the core objective of the research, and the overall structure adheres to conventional academic standards. However, some minor language issues (e.g., typographical errors like "espicially") should be addressed for improved clarity.

Context and Literature Review: The paper provides an extensive literature review, discussing various machine learning and deep learning approaches previously used for malware detection. The work contextualizes the importance of using lightweight models for mobile devices, as well as innovative blockchain integration. However, more detailed comparisons with cutting-edge models from 2023 would further emphasize the novelty of the proposed method.

Figures and Tables: The figures are clear and essential for understanding the flow of the research, especially Figures 1-8 that depict the detection model’s training process and the mobile application interface. Tables provide sufficient details about hyperparameters and model performances, though the paper could benefit from additional visual comparisons with similar models.

Experimental design

Research Scope and Objectives: The research fits within the scope of Android malware detection. The study's key contribution is combining CNN models with blockchain for mobile malware detection, emphasizing real-time efficiency on mobile devices.

Methodological Rigor: The article describes the use of lightweight CNN models (EfficientNetB0, MobileNetV2, and a custom CNN) to achieve Android malware detection. The methodology is sound, but the paper could benefit from a more detailed explanation of how specific parameters (e.g., dropout rates, convolutional layers) were tuned and how blockchain was integrated into the overall workflow.

Reproducibility: The methods section provides sufficient detail to replicate the work, with a clear breakdown of the dataset (CICMalDroid 2020), model architecture, and ensemble methods. However, providing the detailed dataset preprocessing steps, and the exact blockchain configuration would further enhance reproducibility.

Validity of the findings

Model Performance: The reported accuracy of 96.17% from the BlockDroid system is highly competitive compared to the individual CNN models and other prior research. The comparison with existing literature (Table 3) shows a significant improvement in accuracy, which strengthens the validity of the findings.

Innovative Contribution: The combination of lightweight CNN models and blockchain technology is innovative, particularly the concept of storing predictions in a decentralized ledger to prevent repeated analysis. This feature effectively addresses computational resource limitations on mobile devices, adding a unique dimension to the study.

Limitations and Future Directions: While the conclusions are well-supported by the data, the authors could discuss potential limitations more explicitly, such as the reliance on static analysis or the impact of image quality on detection accuracy. Including these limitations and outlining specific future improvements—such as dynamic malware analysis—would strengthen the conclusion.

Additional comments

This research presents a promising approach to Android malware detection using lightweight CNN models and blockchain technology. While the experimental design and findings are robust, further refinement and more explicit discussion of certain methodological choices and limitations could enhance the clarity and impact of the paper.

Reviewer 2 ·

Basic reporting

The manuscript presents "BlockDroid," an innovative Android malware detection system that utilizes lightweight convolutional neural network (CNN) models, ensemble learning, and blockchain technology. By converting Android application files into images, BlockDroid employs CNNs to effectively identify malware patterns. The system enhances detection accuracy and efficiency, achieving a notable accuracy rate of 96.17%. It integrates blockchain technology to securely store detection outcomes, eliminating redundant analyses and optimizing resource use. This approach not only boosts malware detection performance on mobile devices but also ensures the integrity and repeatability of the predictions through decentralized record-keeping.

Experimental design

The choice of the CICMalDroid 2020 dataset is apt for the study's objectives. However, it would be beneficial to provide a more detailed justification for selecting this particular dataset over others. Additionally, discussing any preprocessing steps or the rationale behind the specific distribution of benign versus malware instances could enhance the transparency and replicability of the research.

While the use of ensemble learning and CNNs is well-articulated, the manuscript could benefit from a deeper dive into the configuration of each individual model within the ensemble. Details such as the number of layers, activation functions, and the specific architecture modifications tailored for mobile devices would aid in understanding the robustness and efficiency of the proposed system. I have not seen any innovation in your work.

Validity of the findings

A more comprehensive explanation of the metrics used for performance evaluation (e.g., accuracy, precision, recall, F1 score) and why each is relevant to the context of malware detection would strengthen the manuscript. Additionally, details regarding the division of data into training and testing sets, along with any cross-validation techniques employed, would enhance the credibility of the reported results.

---

## Round 0.2 · Minor Revisions

Dear Authors:

Your manuscript entitled needs some minor revisions before being accepted in PEERJ Computer Science. The comments of the reviewers who evaluated your manuscript are included at the foot of this letter. We ask that you make minor changes to your manuscript based on those comments, before uploading final files.

Thank you for your fine contribution.

Reviewer 2 ·

Basic reporting

The authors have improved the paper, but the related work section still needs a lot of revision. In fact, a critical review has not been done.

Experimental design

N/A

Validity of the findings

N/A

Additional comments

N/A

---

## Round 0.3 · Major Revisions

Dear Authors:

Your manuscript needs some major revisions before being accepted in PEERJ Computer Science. The comments of the reviewers who evaluated your manuscript are included at the foot of this letter. We ask that you make changes to your manuscript based on those comments, before uploading final files.

Thank you for your fine contribution.

Reviewer 2 ·

Basic reporting

Thank you for editing. Still I am not happy with this litrature review section. Literature review is a crucial component of research paper. You must consider these points in your review:

1- Comprehensiveness: It should cover all relevant literature on the topic, including both foundational texts and recent studies. This means including significant studies, theories, and findings that are relevant to the subject. Your manuscript is lack of this point.

2- Scope Definition: Clearly defines the scope of the review, outlining the criteria for inclusion and exclusion of literature. I can not see in your manuscript.

3- Critical Analysis: Goes beyond summarizing sources to critically analyze them. This involves discussing the strengths and weaknesses of different approaches, discrepancies in findings, and gaps in the research. Missed in your paper.

4- Synthesis: Integrates findings from multiple studies to create a cohesive understanding of the topic.

5- Relevance: Demonstrates how each piece of literature contributes to understanding the topic. This involves linking literature findings to the research question or objective of the main study.

6- Current and Referencing: Includes the most recent studies and developments in the field to ensure that the review is up-to-date. There are many papers in 2024 in image based malware detection which you did not consider.

Experimental design

N/A

Validity of the findings

N/A

Additional comments

N/A

---

## Round 0.4 · accepted · Accept

Dear Authors,

Your paper has been revised. It has been accepted for publication in PEERJ Computer Science. Thank you for your fine contribution.

Reviewer 2 ·

Basic reporting

The authors have answered my questions.

Experimental design

The authors have answered my questions.

Validity of the findings

The authors have answered my questions.